# Determinants of cytoplasmic microtubule depolymerization during ciliogenesis in *Chlamydomonas*

Larissa L Dougherty, Prachee Avasthi

At the core of cilia are microtubules which establish length and assist ciliary assembly and disassembly; however, microtubules outside of the cilium can regulate ciliogenesis. The microtubule cytoskeleton polymerizes and depolymerizes rapidly. These processes have been studied across various organisms with chemical and genetic perturbations. However, these have generated conflicting data in terms of the role of cytoplasmic microtubules (CytoMTs) and free tubulin dynamics during ciliogenesis. Here, we look at the relationship between ciliogenesis and CytoMT dynamics in *Chlamydomonas reinhardtii* using chemical and mechanical perturbations. We find that not only can stabilized CytoMTs allow for normal ciliary assembly, but high calcium concentrations and low pH-induced deciliation cause CytoMTs to depolymerize separately from ciliary shedding. In addition, ciliary shedding through mechanical shearing allows cilia to regenerate earlier despite intact CytoMTs. Our data suggest that CytoMTs are not a sink for a limiting pool of cytoplasmic tubulin in *Chlamydomonas*, depolymerization after deciliation is a consequence rather than a requirement for ciliogenesis, and intact tubulin in the cytoplasm and proximal cilium support more efficient ciliary assembly.

## Introduction

Ciliogenesis describes the formation of cilia which are microtubule extensions from the plasma membrane that serve as signaling and sometimes motile components of quiescent cells (Tucker et al, 1979). These largely microtubule-based structures are assembled and maintained through intraflagellar transport (IFT) where tubulin monomers among other components are taken to the tip of the cilium for ciliary assembly and maintenance (Kozminski et al, 1993; Craft et al, 2015). This process is influenced by not only the core proteins required for IFT, but also ciliary gene transcription, protein synthesis, trafficking, and complex preassembly occurring outside of the cilium. All of the required structures and components generate a highly regulated structure and signaling environment within cilia, and because of this, disruptions to any of these

components can lead to various diseases (Reiter & Leroux, 2017). Therefore, it is important to understand how manipulation of cellular processes and components can influence ciliary assembly.

Although cilia are themselves a microtubule superstructure, microtubules are also highly dynamic components elsewhere in the cell. They provide structure and polarity to the cell, serve as highways for molecular trafficking, and facilitate cell division among many other roles (de Forges et al, 2012; Parker et al, 2014). To serve these many roles, tubulin assembles and disassembles very rapidly through the process of dynamic instability which describes how tubulin can disassemble into monomers very quickly from GTP hydrolysis (Mitchison & Kirschner, 1984; Alushin et al, 2014).

Disruption of cytoplasmic microtubule (CytoMT) dynamics has previously been shown to directly impact the ability for cilia to assemble through chemical perturbations. For example, through the use of microtubule inhibitors such as colchicine or colcemid which promote CytoMT depolymerization, cilia cannot assemble at all in Chinese hamster fibroblasts (Stubblefield & Brinkley, 1966) *Chlamydomonas* (Rosenbaum et al, 1969) and *Tetrahymena* (Rosenbaum et al, 1969). Conversely, use of the microtubule stabilizer, taxol, has more recently been shown to inhibit acute ciliary elongation in *Chlamydomonas reinhardtii* (Wang et al, 2013). However, whereas microtubule destabilizing drugs were found to inhibit ciliary assembly, taxol was previously not found to have any effects in female rat kangaroo kidney epithelial cells (PtK1) cells on either assembly or disassembly of primary cilia (Jensen et al, 1987). In contrast, taxol can prevent actin stabilization and microtubule depolymerization-induced ciliary elongation through forskolin, jasplakinolide, and the PKA inhibitor CD. Consistently, the MT depolymerizing drug nocodazole was shown to increase ciliary length likely by freeing up tubulin for ciliogenesis in RPE cells, whereas taxol itself shortened cilia (Sharma et al, 2011). Given these cell-type specific and chemical-specific discrepancies describing the interplay between CytoMT dynamics and ciliary dynamics, it is important to continue investigating how these processes impact one another to better understand how ciliary dynamics are regulated.

CytoMT dynamics with respect to ciliary dynamics have predominantly been studied through chemical perturbations. Here, we explore the requirement for microtubule dynamics in the ciliary

Biochemistry and Cell Biology Department, Geisel School of Medicine at Dartmouth College, Hanover, NH, USA

Correspondence: prachee.avasthi@dartmouth.edu

model organism *C. reinhardtii* with both chemical perturbations and mechanical perturbations to better understand how microtubule dynamics can regulate ciliary assembly. We ultimately find that total CytoMT depolymerization is not required for ciliary assembly.

# Results

## Paclitaxel (PTX)-stabilized CytoMTs do not inhibit ciliary assembly to full length in 2 h

PTX the non-branded Taxol, is a chemical that is well-known for its ability to stabilize microtubules through binding to β-tubulin polymers (Parness & Horwitz, 1981; Manfredi et al, 1982). Stabilizing microtubules with Taxol has previously been found to shorten steady state length in *C. reinhardtii*. In addition, it has been found that through common methods of deciliation in *Chlamydomonas*, CytoMTs depolymerize. These data led to the hypothesis that tubulin used in axonemal assembly for ciliogenesis comes from tubulin present in the cell body (CytoMTs), and microtubule dynamics are required for axonemal microtubule assembly for ciliogenesis (Wang et al, 2013). However, in our previous work, we found that a lower PTX concentration from what has previously been used for ciliary studies can allow for normal ciliary elongation after pH shock in *Chlamydomonas* which raised the question of whether CytoMT dynamics are truly required for ciliary assembly (Dougherty et al, 2023).

We wanted to assess how different PTX concentrations affect mictrotubule stability and ciliogenesis. To test if PTX is indeed stabilizing CytoMTs at lower concentrations in *C. reinhardtii*, we isolated the soluble ("S") and insoluble ("P") β-tubulin fractions after treating cells with 0.5 M acetic acid to induce deciliation ("pH shock") (Witman et al, 1972), 2 mg/ml colchicine, 1% DMSO, 15 $\mu$M PTX, or 40 $\mu$M PTX (Fig 1A). After PTX treatment, both 15 $\mu$M and 40 $\mu$M PTX decreased the presence of β-tubulin in the soluble fraction without completely depleting the pool, confirming that both concentrations sufficiently stabilize CytoMTs in the cells (Fig 1A and B). To directly compare the effect of PTX on ciliogenesis extending beyond steady-state length, cells were treated with increasing concentrations of PTX in the presence or absence of 25 mM lithium chloride (LiCl) (Nakamura et al, 1987). Regardless of PTX concentration, cilia significantly elongated in the presence of LiCl in the CC-1690 WT strain in M1 liquid media (Fig 1C) previously used in Wang et al (2013). We also checked the ability for cilia to completely regenerate from 0 $\mu$m through pH shock in the presence of 15 $\mu$M PTX, 40 $\mu$M PTX, or 60 $\mu$M PTX compared with control cells in DMSO (Fig 1D). Similarly and consistently with our previous work (Dougherty et al, 2023), cilia regenerated normally in 15 $\mu$M PTX, though even at 60 $\mu$M PTX, ciliary length did not significantly differ from DMSO-treated cells at 120 min (Fig 1D). To compare the ability for PTX to stabilize CytoMTs during pH shock, we pretreated cells for 10 min with PTX and then fixed and stained them for β-tubulin at 5 min post pH shock when microtubules are normally depolymerized (Fig 1E). We found that, even immediately after deciliation, CytoMTs are stabilized similarly to pre-deciliated cells as defined by cells which have visible filaments extending past half the long axis of the cell (Fig 1F). Although this population correlation does not

give us changes to tubulin lengths or filament numbers, it provides us with a course measure of how the CytoMT polymerization responds to perturbations. Given that (1) there is already a pool of soluble tubulin at steady state without treatment, (2) decreasing this pool by inducing CytoMT stabilization does not impact ciliogenesis, and (3) increasing this pool by depolymerizing CytoMTs through pH shock also does not impact ciliogenesis, our data suggest that tuning available soluble tubulin access through CytoMT manipulation does not critically impact ciliogenesis.

Whereas deciliation and regrowth of cilia from 0 $\mu$m is coincident with CytoMT depolymerization and stabilization of CytoMTs does not affect cilium regrowth, we wondered whether cilium growth from cilia maintained at steady state showed similar patterns of CytoMT depolymerization. Acute growth from multiple lengths may indicate whether CytoMT depolymerization is specific to cilium growth or deciliation. Using 25 mM LiCl to induce ciliary elongation, we checked the polymerization status of CytoMTs across 30 min during treatment (Fig 2A). Throughout this time, cells were able to elongate ~2 $\mu$m as reported in Wang et al (2013) (Fig 2B). However, upon checking CytoMT polymerization throughout this process, we find that these microtubule arrays remained undisturbed in both untreated and LiCl-treated cells (Fig 2C). These data suggest that CytoMT depolymerization is not necessary for ciliary assembly, though it is possible that finer CytoMT depolymerization could occur on a level that is not detectable with these methods.

## Chemically induced ciliary shedding occurs separately from CytoMT depolymerization

Given that CytoMT depolymerization does not occur after ciliary elongation from steady state lengths (Fig 2) and cilia assemble to full length in 2 h with an available pool of soluble tubulin despite CytoMT stabilization (Fig 1), we suggest that detectable CytoMT depolymerization through pH shock is not a requirement for ciliary assembly. To better understand if depolymerization is a result of physical loss of the cilium versus other factors coincident with cilium loss, we utilized a previously characterized *fa2-1* mutant encoding a mutant NIMA family kinase which cannot deciliate in the presence of calcium or acid (Mahjoub et al, 2002) (Fig 3A). After pH shock, we found that whereas WT cells shed 100% of their cilia and depolymerized ~70% of their CytoMTs (Fig 3B, left), the *fa2-1* mutants maintained cilia as expected, but also depolymerized CytoMTs similar to WT (Fig 3B, right). Despite loss and reformation of CytoMTs, ciliary length remained unchanged (Fig 3C). This is surprising because it was previously reported that CytoMT depolymerization is required for ciliogenesis (Wang et al, 2013), but here we see that even in the absence of deciliation and during pH shock, CytoMTs still depolymerize. This suggests that CytoMT depolymerization could be an artifact or consequence of pH shock and is not strictly tied to loss of cilia.

To test if CytoMT depolymerization was specific to pH shock or if other agents that induce ciliary shedding can also induce CytoMT depolymerization separately, it has previously been established that pH shock induces a calcium influx into the cell (Quarmby, 1996), and introducing the cell to higher concentrations of calcium can induce ciliary shedding (Quarmby & Hartzell, 1994). To further test what triggers CytoMT depolymerization coincident with deciliation, we tested if calcium influx, known to occur upon pH shock, was also

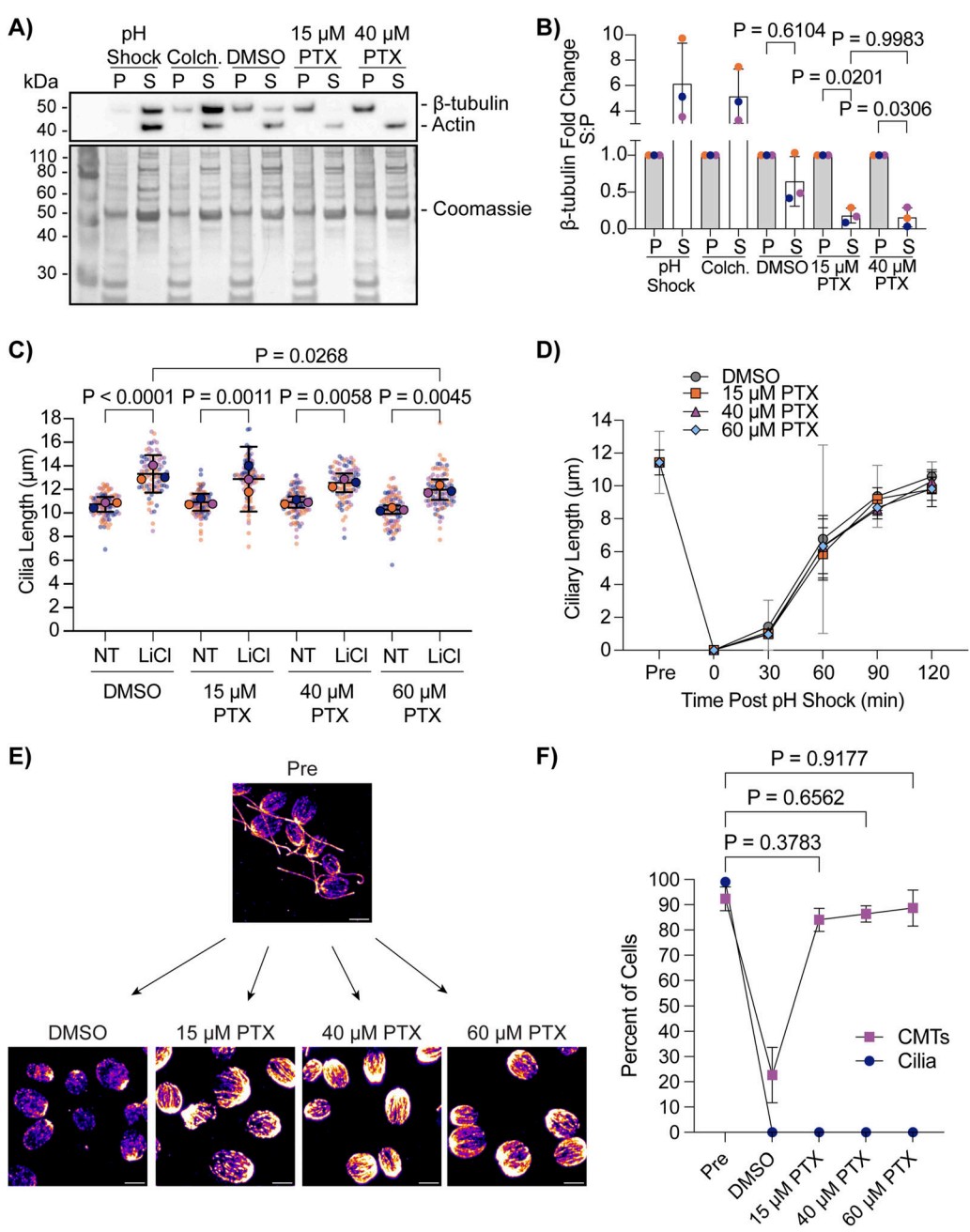

**Figure 1. Paclitaxel-stabilized CytoMTs permit normal ciliogenesis.**
**(A)** WT cells (CC-5325) were treated with acetic acid, 2 mg/ml colchicine for 100 min, 1% DMSO for 10 min, 15 µM paclitaxel (PTX) for 10 min or 40 µM PTX for 10 min. Protein was collected and probed for β-tubulin, actin (soluble protein control) or total protein with Coomassie on a PVDF membrane. **(A, B)** Quantification of Western blot band intensity in (A). Microtubule intensity was normalized to total protein and then the soluble tubulin (S) intensity was compared with the insoluble intensity (P). Significance was determined using a one-way ANOVA with Šidák's multiple comparisons test (N = 3). Error bars are mean with SD. For all tests, P < 0.05 is considered significantly different. **(C)** WT cells (CC-1690) were grown in M1 media overnight and then treated with 1% DMSO or increasing concentrations of PTX with or without 25 mM LiCl for 30 min. Significance was determined with a one-way ANOVA and Šidák's multiple comparisons test (n = 30, N = 3). Orange circles are trial 1, purple circles are trial 2, and navy circles are trial 3. Small circles represent individual data points and overlayed large circles indicate the mean for the trial. **(D)** WT cells (CC-5325) were grown in TAP media overnight, pretreated in increasing concentrations of PTX for 10 min, and then regenerated for 2 h after pH shock with acetic acid and PTX. Significance was determined with a one-way ANOVA and Šidák's multiple comparisons test (n = 30, N = 2). Error bars are mean with 95% confidence interval. **(D, E)** Representative images of β-tubulin-stained cells at the 5-min time point post pH shock in (D). Scale bars are 5 µm. **(F)** Quantification of the percent of cells with cilia and polymerized cytoplasmic microtubules (CytoMTs) at 5 min post pH shock in DMSO or PTX. Navy blue circles are cilia; purple squares are CytoMTs. Error bars are mean with SD (n = 100, N = 3). Significance for CytoMTs is represented on the graph and determined using a one-way ANOVA with Šidák's multiple comparisons test. Cells considered to have polymerized microtubules have visible β-tubulin containing filaments extending past half of the cell.

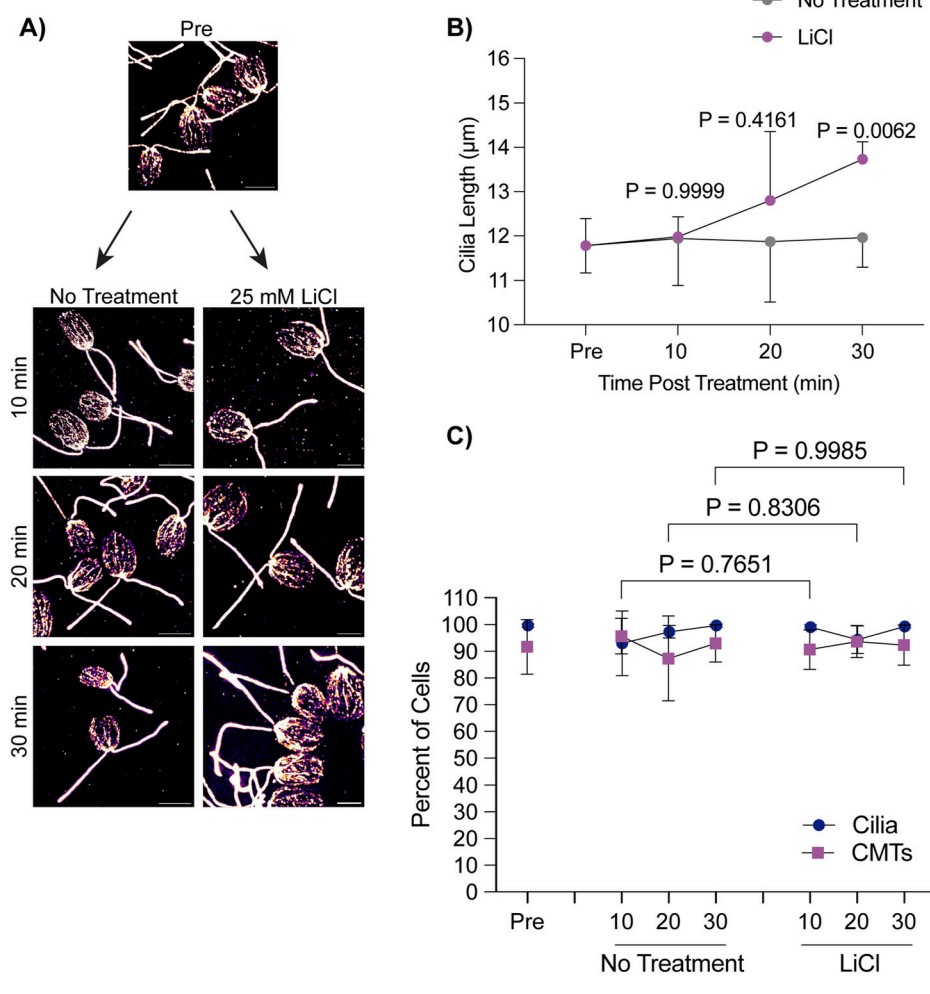

**Figure 2. Lithium chloride-induced ciliary elongation does not require CytoMT depolymerization from steady state.**
**(A)** WT cells were treated with either 25 mM LiCl or no treatment for 30 min and then fixed and stained for $\beta$-tubulin. Scale bars are 5 $\mu$m.
**(A, B)** Quantification of cilia length for the experiment described in (A). Error bars are mean with 95% confidence interval (n = 30, N = 3). Significance compares cilia length at 30 min between treatments and was determined with a two-way ANOVA with Šídák's multiple comparisons test.
**(A, C)** Quantification of percent of cells with cilia versus polymerized microtubules for the experiment described in (A). Error bars are mean with SD (n = 100, N = 3). Significance for CytoMTs is represented on the graph and determined using a one-way ANOVA with Šídák's multiple comparisons test.

sufficient to induce CytoMT depolymerization. We introduced cells to 75 mM calcium chloride ("CaCl$_2$") in 5 mM HEPES ("Buffer") for 10 min which is long enough to induce robust ciliary shedding. After ciliary shedding, cilia regenerated similar to cilia exposed to pH shock (Fig 4A). We then measured the effect of calcium-induced ciliary shedding on CytoMT stability. Upon ciliary shedding, we found that CytoMTs also depolymerized quickly and then reformed similar to pH shock (Fig 4B). To determine if calcium has the same effect on ciliated cells, we repeated this experiment in *fa2-1* mutants (Fig 4C–E). Similar to WT cells, *fa2-1* mutants also exhibited depolymerized microtubules though they did not depolymerize to the same degree as WT (Fig 4E). These data indicate that depolymerization is specific to calcium-mediated mechanisms rather than cilia loss.

## Mechanically sheared cilia do not depolymerize CytoMTs and regenerate cilia faster than following chemical shearing

Next, we wanted to assess changes to CytoMT polymerization through nonchemical means of ciliary loss. We introduced the cells to a cell homogenizer/bead beater for 3 min to excise cilia and then looked at CytoMTs as cilia regenerated (Fig 5A). Interestingly, mechanical shearing did not induce robust CytoMT depolymerization (Fig 5B). Furthermore, mechanically sheared cilia were able to regenerate back to full length much faster (in 60 min), whereas cilia shed via pH shock were only able to regenerate to half-length in this time (Fig 5C). Looking more closely at regeneration in these processes, we found that mechanically sheared cilia, which maintain a ~1 $\mu$m ciliary track immediately after ciliary excision, can immediately begin steadily regenerating overtime (Fig 5C). In contrast, pH-shocked cells have delayed ciliogenesis, but after this process is started, cells can more rapidly regenerate cilia (Fig 5B and C). To confirm that this was not a consequence of CytoMT reestablishment, we compared ciliogenesis with CytoMT polymerization. Though pH-shocked cells repolymerized CytoMTs within 10–20 min, ciliogenesis did not continue until 30–45 min post deciliation unlike mechanically sheared cells which maintained both unperturbed CytoMTs and cilia throughout (Fig 5B and C). These data show that in the case where cilium growth happens faster, CytoMT depolymerization does not occur. Our data may

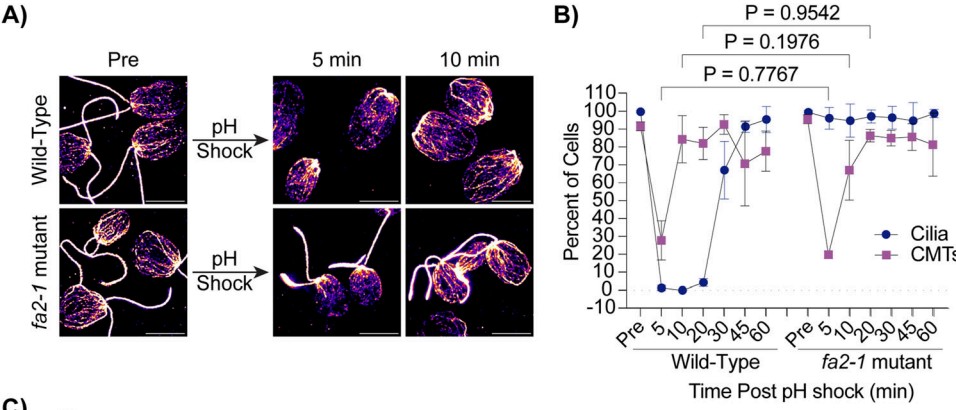

**Figure 3. CytoMT depolymerization occurs separately from ciliary shedding after pH shock.**
**(A)** WT or *fa2-1* mutants were pH shocked with acetic acid for 45 s and then regrown in fresh TAP for 60 min. Cells were fixed and stained for β-tubulin. Scale bars are 5 μm. **(B)** Quantification of percent of cells with cilia or CytoMTs post pH shock. Error bars are mean with SD (n = 100, N = 3). Significance was determined with a one-way ANOVA and Šídák's multiple comparisons test for comparisons between CytoMTs. **(A, C)** Superplots of ciliary length measured from *fa2-1* cells in (A). Error bars are mean with 95% confidence interval (n = 30, N = 3). Significance was determined with a one-way ANOVA (*P* = 0.5573).

suggest that polymerized CytoMTs facilitate more rapid ciliogenesis. Alternatively, the existence of cilium initial segments may promote more rapid assembly completely independent of CytoMT state that is dependent instead upon calcium influx (Fig 4) that may not be triggered upon mechanical shearing. It could be that initiation of ciliary assembly is rate limiting.

The delay in ciliary shedding because of pH shock may differentially affect CytoMTs based on modifications that add stability. One microtubule modification, acetylation, provides stability, strength, and flexibility to microtubules (Janke & Montagnac, 2017). It is possible that during pH shock, changes to acetylation can occur that may be the cause for the delay in ciliogenesis. To test this, we also compared the presence of acetylated α-tubulin in pH-shocked cells versus mechanically sheared cells. Deacetylation of α-tubulin has been found to work in concert with ciliary disassembly during mitosis, and acetylation can act as a signal for downstream events such as cell differentiation in adipocytes (Forcioli-Conti et al, 2016). Acetylation recruits further modifications to tubulin that allow for motility and ciliary length maintenance in *Chlamydomonas* and human cells (Tripathi et al, 2021). It is possible that pH shock could be inducing upstream pathways that inhibit CytoMT acetylation must be maintained for ciliogenesis. Comparing pH shock with mechanical shearing, we found that during pH shock, acetylated CytoMTs were significantly shorter both overall (Fig 5D and E) and when comparing the longest acetylated CytoMT per cell (Fig 5D and F). These data show that pH shock may affect both stabilized and non-stabilized CytoMTs or affect the degree of acetylation of the stabilized population.

Delayed ciliogenesis upon microtubule depolymerization could indicate that there is a recruitment defect for proteins necessary for ciliogenesis on the cytoplasmic tracks needed for this recruitment or local organization. Using a previously tagged mScarlet-IFT54 anterograde IFT subunit and sfGFP-tagged retrograde IFT140 subunit generated and validated by Wingfield et al (2021), we compared fluorescence of these complexes at the ciliary base during pH shock and mechanical shear (Fig 6A and C). We found that during pH shock, there is a significant and more robust increase in fluorescence at the base of cilia for both anterograde (Fig 6A and B) and retrograde (Fig 6C and D) trafficking complexes during pH shock as compared with mechanical shearing which remains increased at 10 min when CytoMTs have reformed (Fig 3B). The mitigation of the robust increase in mechanically sheared cells may be because of the availability of a ciliary sink as the cilia are more rapidly growing in this case. In other words, these data suggest 2 potential factors that impact the quantity of IFT proteins at the ciliary base: (1) the availability of CytoMTs for organization or recruitment of IFT proteins and (2) the presence of a fast-growing cilium with an existing ~1 μm track that acts as a sink for IFT proteins.

## Discussion

Throughout this work, we identify the relationship between ciliogenesis and CytoMT polymerization which we tease apart by holding either of these processes constant while varying possible determinants. We find that although CytoMT depolymerization does occur during deciliation with some techniques commonly used to induce deciliation, these dynamics are not necessarily required for ciliary assembly and instead may inhibit ciliary assembly because

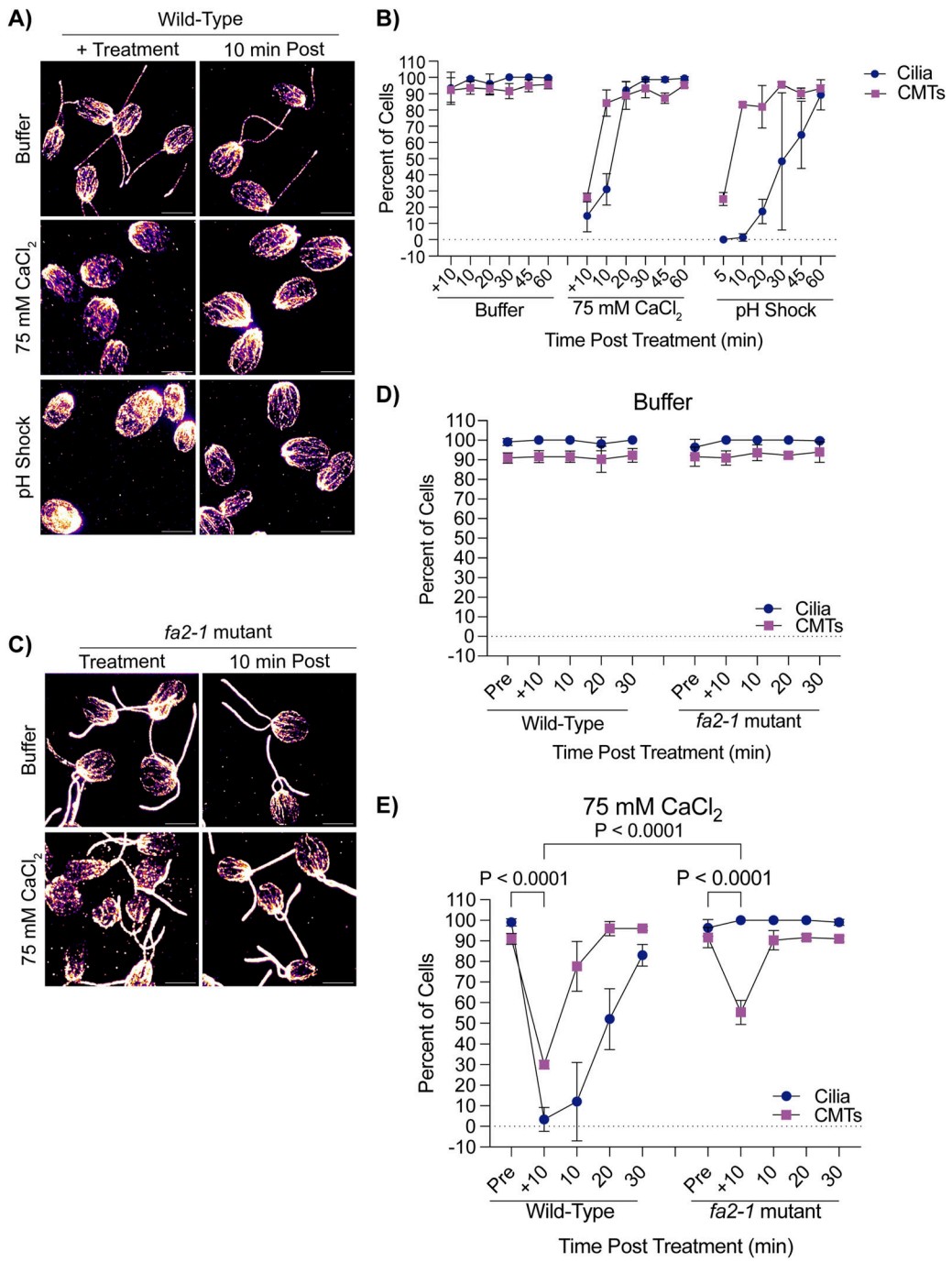

**Figure 4. CytoMT depolymerization occurs separately from ciliary shedding during high calcium.**
**(A)** WT cells were either pretreated for 10 min in 5 mM HEPES ("Buffer"), 10 min in 75 mM CaCl$_2$ or 45 s in acetic acid, and then fixed and stained for β-tubulin. Scale bars are 5 μm. **(A, B)** Quantification of cilia and CytoMTs from (A). Error bars are mean with SD (n = 100, N = 3). **(C)** Representative images of *fa2-1* mutants after treatment in buffer or 75 mM CaCl$_2$. Cells were fixed and stained for β-tubulin. Scale bars are 5 μm. **(C, D, E)** Quantification of WT (CC-5325) or *fa2-1* mutants in buffer (C) or 75 mM CaCl$_2$ (E). **(C)** Error bars are mean with SD (n = 100, N = 3) for the experiment described in (C). Statistics compare CytoMTs and were determined using a one-way ANOVA and Šídák's multiple comparisons test.

of CytoMT depolymerization as we see a reestablishment in acet-ylated microtubule length before ciliogenesis can begin (Fig 5E and G) and a need for IFT recruitment to the ciliary base (Fig 6). These findings are summarized in Fig 7. Our data are consistent with a

model in which events happening during deciliation have effects on associated CytoMT processes, but these are not required for cil-iogenesis. These data shed light on two important points: (1) tubulin does not necessarily need to be freed up from the cell to allow

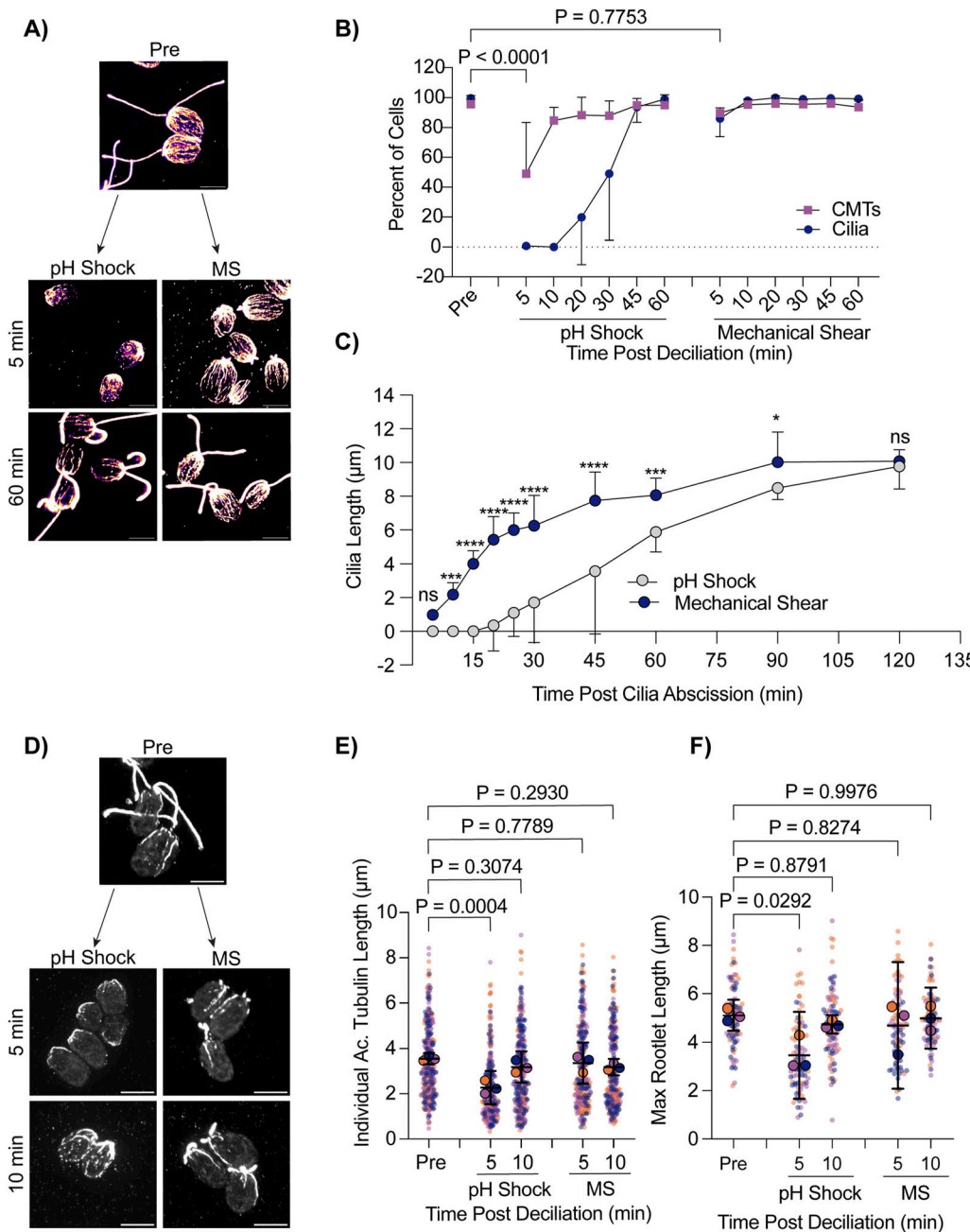

**Figure 5. Cells maintain intact CytoMTs during mechanical shearing and regenerate cilia more quickly.**
**(A)** WT cells were either pH-shocked for 45 s or placed in a cell homogenizer ("Mechanical Shear", "MS") for 3 min. Representative images are cells fixed and stained for β-tubulin. Scale bars are 5 μm. **(B)** Quantification of percent of cells with cilia (navy circles) compared with polymerized microtubules (purple squares) between pH shock (left) and MS (right). Error bars are mean with SD. Statistics compare CytoMTs. Significance was determined using a one-way ANOVA and Šídák's multiple comparisons test (n = 30, N = 3). **(C)** Quantification of cilia length over 2 h. Error bars are mean with 95% confidence interval (n = 30, N = 3). Statistics were determined using a two-way ANOVA with Šídák's multiple comparisons test (n = 30, N = 3). (ns - P > 0.05, *P ≤ 0.05, **P ≤ 0.01, ***P ≤ 0.001, ****P ≤ 0.0001). **(D)** WT cells were fixed and stained for acetylated α-tubulin after pH shock or MS. Scale bars are 5 μm. **(E, F)** Quantification of individual (E) or max (F) acetylated microtubule lengths. Error bars are mean with 95% confidence interval. Statistics were determined using a one-way ANOVA and Dunnett's multiple comparisons test (n = 30 cells, N = 3).

ciliary assembly to occur and (2) solubilization of cytoplasmic tubulin is neither required nor does it block the mobilization of the intracellular pool of ciliary protein precursors which we see through stimulated IFT recruitment both during pH shock and mechanical shearing, though this response is less robust during mechanical shearing.

Whereas this work addresses the requirement and assembly of tubulin in CytoMTs for ciliary dynamics, these experiments are

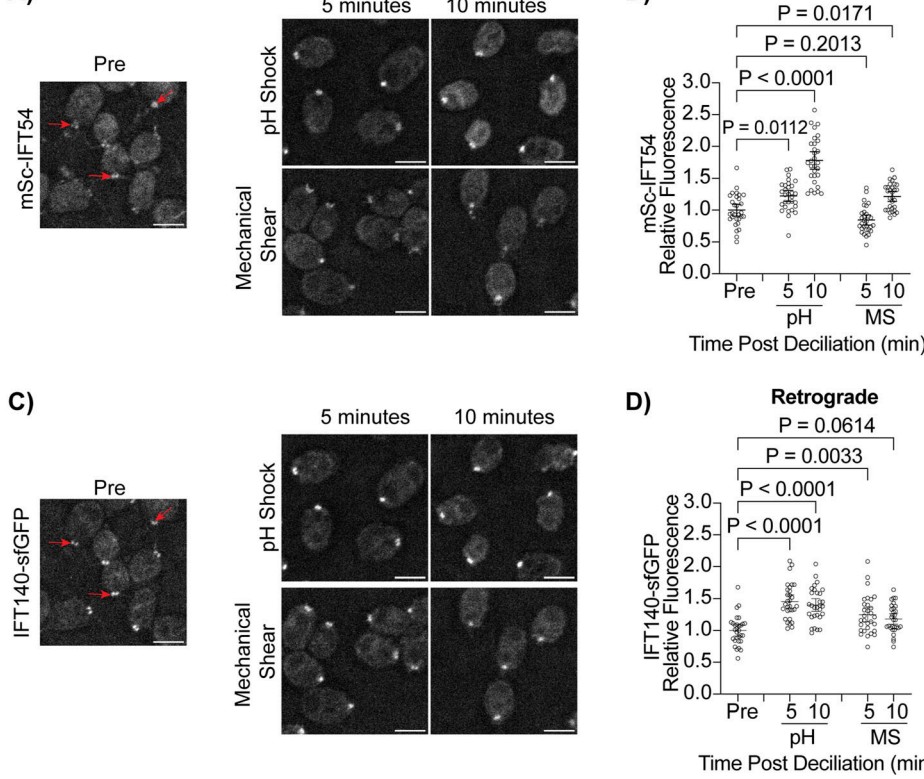

**Figure 6.  PH shock-induced ciliary shedding increases IFT recruitment to the ciliary base.**
**(A, C)** mSc-IFT54/IFT140-sfGFP were fixed and imaged. Fluorescence at the base of the cilium was quantified (red arrows). **(A, C)** Representative images show cells pre deciliation ("Pre"), 5 min, and 10 min post treatment for both pH shock and mechanical shear for both mSc-IFT154 (A) and IFT140-GFP (C) transport. Scale bars are 5 $\mu$m. **(B, D)** Quantification of mSc-IFT54 (B) or IFT140-sfGFP (D) fluorescent intensity at the ciliary base. Error bars are mean with 95% confidence interval (n = 30, N = 1). Statistics were determined using a one-way ANOVA and Šídák's multiple comparisons test.

under conditions in which a soluble tubulin pool remains given that there is still a measurable amount of tubulin present both with 15 $\mu$M PTX and 40 $\mu$M PTX (Fig 1A and B). It is possible that this amount of tubulin is enough to supply the start of new cilia while the cell begins generating new tubulin for the assembling cilium which occurs immediately after deciliation (Albee et al, 2013). Soluble tubulin that is not incorporated into tubulin filaments during PTX treatment could be degraded or sequestered, preventing it from binding to other microtubule filaments. It is also possible that the cell homeostasis requires maintenance of some level of soluble tubulin. It has been shown that taxol can induce an increase in tubulin synthesis which makes it likely that the cell can maintain a level of soluble tubulin which can saturate Taxol in the cell and give rise to a constant pool of soluble tubulin (Stargell et al, 1992).

Effects of pH shock and calcium on live microtubule dynamics alone have previously been investigated (Liu et al, 2017); however, the stability of existing microtubules has not been fully explored. Liu et al (2017) found that, by using mNeonGreen-tagged EB1, a microtubule end-binding protein found on the growing end of microtubules, microtubule growth was frozen in *Chlamydomonas* by decreasing pH and increasing calcium concentration. They also noted that the different *Chlamydomonas* tubulin isoforms have different isoelectric points ($\alpha$-tubulin = 5.01, $\beta$-tubulin = 4.82, and EB1 = 5.7), so dropping the pH below these values which allows for sufficient calcium influx to induce ciliary shedding is also sufficient to separately induce CytoMT depolymerization, along with the ability for calcium alone to inhibit microtubule dynamics. However,

despite the effect of pH on CytoMT stability shown in this work, there is also a considerable lag between the time that CytoMTs are reestablished and the timing of initial ciliogenesis (Figs 3B and 5B) which raises the possibility that other signaling pathways that influence ciliogenesis are inhibited during this process. For example, it was previously found that pH shock activates the microtubule depolymerizing kinesin CrKin13 which has also been implicated in ciliogenesis (Wang et al, 2013).

In addition, this lag in ciliogenesis after pH shock could be because of a lack of CytoMTs altogether for recruitment or organization of proteins needed for ciliary assembly. The well-studied agglutinin protein, SAG1, is one protein which must move from the plasma membrane to the ciliary base upon induction. It has been found that this movement occurs on CytoMTs through the retrograde transport motor cytoplasmic dynein 1b, and after this movement, it can diffuse into the peri-ciliary membrane independent of IFT (Belzile et al, 2014; Cao et al, 2015; Ranjan et al, 2019). In addition, other ciliary proteins have been found to be assembled in the cytoplasm before reaching the cilium including the cytoplasmic dynein arm 1I (Viswanadha et al, 2014) and channelrhodopsin which move on acetylated rootlet microtubules through IFT to reach the eyespot (Mittelmeier et al, 2011; Awasthi et al, 2016). During pH shock-induced CytoMT depolymerization, it is possible that without the cytoplasmic highways, proteins cannot reach the ciliary base for transport into the cilium and this is one factor by which ciliogenesis could also be stalled in addition to the delayed build-up of IFT at the ciliary base after 10 min (Fig 6B) when microtubules are repolymerized (Fig 4A). Future work needs to be

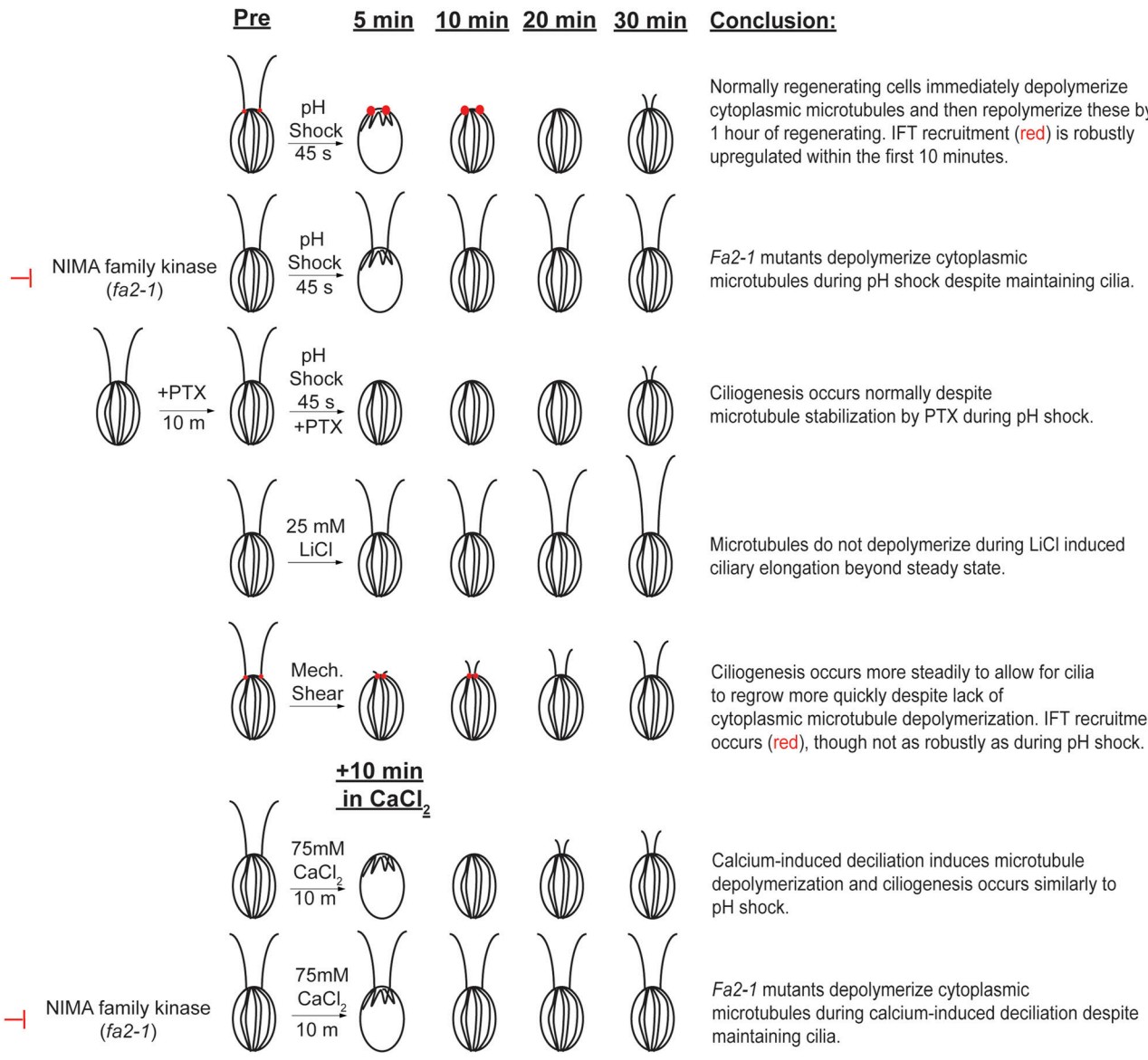

**Figure 7. Summary of microtubule dynamics in conjunction with ciliogenesis.**
Results from each trial are summarized for each of the different conditions tested up to 30 min. Red dots represent IFT fluorescence tested for both pH shock (top) and mechanical shear (bottom). Adapted from Dougherty et al (2022) *Preprint*.

done to better understand how different methods of deciliation can ultimately impact cellular processes that directly regulate ciliogenesis and what molecular factors regulate the entry of proteins already at the base during pH shock-induced deciliation versus other mechanisms of deciliation.

# Materials and Methods

## Strains and maintenance

The WT strains (CC-5325 and CC-1690), *fa2-1* mutant (CC-3751), and IFT strain (ift140-1::IFT140-sfGFP ift54-2::mS-IFT54 mt; CC-5862) were acquired from the *Chlamydomonas* Resource Center. Cells were maintained on 1.5% Tris–acetate–phosphate (TAP) plates under constant light. For experiments, CC-1690 cells were inoculated into liquid M1 media, and CC-5325 cells, *fa2-1* mutant cells, and IFT cells were inoculated in TAP and grown overnight under constant light and agitation. All media recipes were used according to the *Chlamydomonas* Resource Center.

## Ciliary length experiments

### *Steady-state ciliary length experiments*
Overnight cultures were resuspended in fresh TAP or M1 (CC-1690) the next morning. Cells were treated with either DMSO, PTX (P3456; Invitrogen), colchicine (Sigma-Aldrich), or LiCl (Sigma-Aldrich), fixed in equal amounts of 2% glutaraldehyde (Electron Microscopy

Sciences), and then imaged on a Zeiss Axioscope 5 DIC with 40x magnification and Zeiss Zen 3.1 software.

### PH shock-induced ciliary shedding and regeneration

Overnight cultures were resuspended in fresh TAP and pretreated or not with PTX 10 min before deciliation. To induce deciliation, 0.5 M acetic acid was added to the cell suspension to bring the pH down to 4.5 for 45 s, and then brought back up to 7.0 with 0.5 M potassium hydroxide. Cells were then spun down at 600$g$ for 1 min and resuspended in new TAP with or without PTX.

### Calcium chloride-induced ciliary shedding and regeneration

Overnight cultures resuspended in fresh TAP were spun down for 60 s at 600$g$ at room temperature and resuspended in either 5 mM HEPES or 75 mM calcium chloride in 5 mM HEPES for 10 min. Then, the cells were resuspended in fresh media and allowed to re-generate cilia.

### Mechanical shear-induced ciliary shedding and regeneration

Overnight cultures resuspended in fresh TAP were placed in a cell homogenizer (Analog Disruptor Genie) for 3 min, spun down for 60 s at 600$g$ at room temperature, and then resuspended in fresh TAP to regenerate.

### Immunofluorescence and quantification

#### β-Tubulin

Staining was performed as described in Wang et al (2013). Briefly, cells were adhered to poly-Lysine–coated coverslips, fixed in 30 mM HEPES (pH = 7.2), 3 mM EGTA, 1 mM MgSO$_4$, and 25 mM KCl with 4% PFA for 5 min, permeabilized with 0.5% NP-40 for 5 min, and then fixed in ice-cold methanol for 5 min. Coverslips were blocked in 5% BSA and 1% Fish ("Block") gelatin for 30 min, then in block with 10% Normal Goat Serum (Sigma-Aldrich), and then incubated in β-tubulin primary antibody (2146S; CST) diluted in 20% block in PBS overnight at 4°C. Coverslips were washed 3 × 10 min in PBS, incubated in secondary antibody (Alexafluor 488 goat anti-rabbit IgG, A11008, 1:500) for 1 h, washed 3 × 10 min in PBS, and allowed to dry. Coverslips were mounted in Fluoromount G (Thermo Fisher Scientific) before imaging. Cells were imaged on a Nikon Yokogawa SoRa super resolution spinning disk confocal with 100x oil objective (data collection) and 2.8x magnifier (representative images). For quantification, max intensity projections were generated from z stacks and 100 cells per time point were counted using the Cell Counter tool in FIJI for polymerized or depolymerized CytoMTs. Cells considered to have polymerized CytoMTs were cells with fluorescent CytoMTs spanning more than half the long axis of the cell (ciliary apex to the cell base). Cells considered to have depolymerized CytoMTs were cells with fluorescent CytoMT filaments that spanned less than half the long axis of the cell. These typically barely extended into the cell from the ciliary apex. The presence or absence of cilia was noted alongside the CytoMT assessment per cell.

#### Acetylated α-tubulin

Cells adhered to coverslips were fixed in 4% PFA in 10 mM HEPES for 15 min, placed in ice-cold 80% acetone for 5 min, placed in ice-cold 100% acetone for 5 min, and then allowed to dry before rehydrating for 5 min in PBS. Then cells were blocked in 100% Block for 30 min, 10% Normal Goat Serum, and stained for acetylated tubulin (1:1,000) overnight at 4°C. Coverslips were washed 3 × 10 min in PBS incubated with secondary (Alexa Fluor 499 goat anti-mouse IgG, A11001; Invitrogen) for 1 h at RT, covered, and then washed 3 × 10 min in PBS, covered before allowing to dry completely and mounted with Fluoromount G. For quantification, max IPs were generated and acetylated microtubules were measured using FIJI's segmented line tool starting at the apex of the cell where acetylated microtubules originate and following along the length of the continuous acetylation signal for 30 cells per timepoint.

#### IFT140/54

Cells adhered to coverslips were placed in ice-cold 100% methanol 2 × 5 minutes, 50% methanol in PBS for 5 min, then PBS for 5 min before allowing to dry, mount, and image. For quantification, z stacks were summed together, background subtracted with rolling ball radius = 50, and then equal-sized circles drawn over the signal at the base of the cilium. Fluorescent intensity was calculated using the calculation for CTCF as previously described (Dougherty et al, 2023).

### SDS–PAGE and immunoblotting

Soluble (S) and insoluble (P) fractions were collected similarly to Wang et al (2013). Briefly, 1 ml of treated cells were collected by centrifugation at 600$g$ for 1 min at RT and resuspended in 50 $\mu$l TMMET Buffer (20 mM Tris–HCl [pH 6.8], 0.14 M NaCl, 1 mM MgCl2, 2 mM EGTA, 4 $\mu$g/ml Taxol, and 0.5% Nonidet P-40 Substitute) with 2 × 30 s of vigorous pipetting and 2 × 30 s of gentle vortexing (half-max vortex speed). Cells sat for 10 min with one gentle round of vortexing at 5 min and at 10 min before spinning down at 21,000$g$ for 10 min at RT. The supernatant was collected ("S" fraction) and then the pellet was washed with TMMET buffer. Finally, the remaining pellet was resuspended in 100 $\mu$l TMMET buffer ("P" fraction). For loading protein samples into the gel, the S fractions were diluted 1:14 in TMMET buffer and P fractions were diluted 1:56 in TMMET buffer (determined according to detectable signal within the linear range of the antibodies after blot imaging). Protein from S and P fractions were mixed with 2 $\mu$l 0.5 M DTT and 4x LDS (Invitrogen), incubated at 70°C for 10 min and run on NuPAGE 10% Bis-Tris gels in MES/SDS–PAGE running buffer (1 M MES, 1 M Tris base, 69.3 mM SDS, 20.5 mM EDTA free acid). Protein was transferred onto PVDF membrane, blocked in 5% milk (wt/vol, Signature Select Instant Nonfat Dry Milk) in PBST (1x PBS, 0.1% Tween20), incubated in β-tubulin (1:1,000, 2146S; CST) or actin C4 (1:1,000, MAB1501; Sigma-Aldrich) diluted in 1% milk with 1% BSA in PBST overnight at 4°C. Blots were washed 3 × 10 min in PBST, incubated in secondary antibody diluted in 1% milk with 1% BSA (1:5,000, G21234; Invitrogen; 1:5,000, 31430; Invitrogen) for 1 h at RT. Blots were washed 3 × 10 min in PBST and then incubated with Pico Chemiluminescent substrate (Invitrogen) and imaged on the Syngene GBOX CHEMI XR 5 using GeneSys software.

### Statistical analysis

Data were collected and organized in Excel, and then graphed and analyzed with GraphPad Prism Version 9.5.1 (528). SuperPlots were

made and analyzed according to Lord et al (2020) where statistics were performed on the averages of three trials and individual trials are represented by corresponding symbol colors. For experiments, "n" is the number of cells quantified and "N" is the number of experiments. Small circles are individual data points. Large circles are averages for each trial. Orange circles represent trial 1 data, purple circles represent trial 2 data, and navy circles represent trial 3 data.

## Data Availability

This study includes no data deposited in external repositories.

## Supplementary Information

## Acknowledgements

We would like to thank the Barlowe Laboratory at Dartmouth College and Brae Bigge at Arcadia Science for their feedback on experimental ideas and revising the article. We would like to thank Karl Lechtreck for his close reading of the article and catching an error in mutant identification. We would like to thank the reviewers for their constructive suggestions to enhance the clarity of the article. We would also like to thank Ann Lavanway at the Life Sciences Light Microscopy Facility at Dartmouth for help with microscopy. We would also like to thank the Biochemistry and Cell Biology Department at Dartmouth College for providing resources to complete this work. This work was funded by NIH/NIGMS MIRA R35GM128702 (P Avasthi).

### Author Contributions

LL Dougherty: conceptualization, formal analysis, validation, investigation, and writing—original draft, review, and editing.
P Avasthi: conceptualization, supervision, funding acquisition, and writing—review and editing.

### Conflict of Interest Statement

P Avasthi is the CSO at Arcadia Science.

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
