## [Reviewer comments · Life Science Alliance]

Life Science Alliance

Determinants of cytoplasmic microtubule depolymerization during ciliogenesis in *Chlamydomonas*

Larissa Dougherty and Prachee Avasthi

DOI: <https://doi.org/10.26508/lsa.202302287>

Corresponding author(s): Prachee Avasthi, Geisel School of Medicine at Dartmouth College

Review Timeline:

Submission Date:	2023-07-21
Editorial Decision:	2023-08-18
Revision Received:	2023-09-15
Editorial Decision:	2023-09-19
Revision Received:	2023-09-29
Accepted:	2023-10-03

Transaction Report:

August 18, 2023

Re: Life Science Alliance manuscript #LSA-2023-02287

Dr. Prachee Avasthi
Geisel School of Medicine at Dartmouth College
74 College St
HB 7200
Hanover, NH 03755

Dear Dr. Avasthi,

Thank you for submitting your manuscript entitled "Determinants of cytoplasmic microtubule depolymerization during ciliogenesis in Chlamydomonas" to Life Science Alliance. The manuscript was assessed by expert reviewers, whose comments are appended to this letter. We invite you to submit a revised manuscript addressing the Reviewer comments.

Thank you for this interesting contribution to Life Science Alliance. We are looking forward to receiving your revised manuscript.

Sincerely,

B. MANUSCRIPT ORGANIZATION AND FORMATTING:

Reviewer #1 (Comments to the Authors (Required)):

Chlamydomonas is a great model for studying mechanisms of ciliogenesis. How cilia regenerate, and how cell's utilize their tubulin resources to grow or rebuild shed cilia are questions that apply across species and potentially to disease states. One might suppose that depolymerized tubulin is required to have free tubulin on hand to build or re-build cilia, but the authors have conducted a series of experiments and provide compelling evidence that is not the case. It is also surprising how different modes of cilia shedding and regeneration rates are not exactly correlative to recruitment of IFT components to the cilia base.

The manuscript is well written and data/analysis is rigorous. I only had minor comments that should be addressed.

Fig. 5B. I am confused or maybe there was some mistake in 5B (mechanical shear)? The data for cilia (blue circles) shown on the right suggest cilia are not lost after mechanical shear. However that is opposite of what you see in Fig 5A and C in the same timeframe. Please reconcile.

Fig 5E,F. Looks like individual data points are in ~3 different colors. Is the coloring important or should there be a legend? As is, it is a little hard to see the mean/error bars.

Fig S1. This seems pretty important point to the paper and is even referred to twice in the discussion. Why not make it Fig. 6? It seems like an interesting but rather counter-intuitive point that cells that are slower to regenerate cilia are actually more rapidly/intensely recruiting IFT components. It was discussed reasonably but I felt this should be a little more front and center in the MS rather than supplemental.

Fig S2. I also felt like it should be a main figure. There are a lot conditions to digest and it may be helpful to the reader to have this up front for comparison rather than having to re-direct.

Line 125 - please define PtK1 cells.

Reviewer #2 (Comments to the Authors (Required)):

Microtubules play an important role in ciliogenesis. While many researchers have indicated that microtubules are a key component of cilia, it is unknown how cytoplasmic microtubules contribute to ciliogenesis. The authors investigate the link between ciliogenesis and cytoplasmic microtubule regulation in Chlamydomonas using chemical (Paclitaxel) and mechanical disruption techniques. They discovered that (1) the availability of soluble tubulin via cytoplasmic microtubules is not essential for ciliogenesis, (2) cytoplasmic microtubule depolymerization is not required for cilia construction, and (3) rather intact cytoplasmic microtubules significantly facilitate ciliogenesis. This work is well-written, and overall the findings are convincing. I just have one question. The observation and measurement of cytoplasmic microtubules would be critical in this article to validate the author's hypothesis. Except for Fig. 1A, it is unclear how the authors quantify β -tubulin. In Figs. 3 and 4, for example, how did the authors investigate cilia and cytoplasmic microtubules positive Chlamydomonas? Could authors simply differentiate (or measure) cilia+/CMTs- and/or cilia-/CMTs+? In cilia+ cells, was cilia length comparable between cilia+/CMTs- cells? These must be clarified before publishing. In summary, this work will be useful for understanding Chlamydomonas ciliogenesis, and this information may also be critical to understanding ciliogenesis in animal cells.

We thank the reviewers for their examination of the manuscript. We have revised the manuscript accordingly and addressed all of the reviewers comments below. We have expanded on our methods for clarity in the manuscript, quantified ciliary lengths in *fa2-1* mutant cells pre and post pH shock, and added minor changes for text clarity and figure clarity.

Reviewer #1 (Comments to the Authors (Required)):

Chlamydomonas is a great model for studying mechanisms of ciliogenesis. How cilia regenerate, and how cell's utilize their tubulin resources to grow or rebuild shed cilia are questions that apply across species and potentially to disease states. One might suppose that depolymerized tubulin is required to have free tubulin on hand to build or re-build cilia, but the authors have conducted a series of experiments and provide compelling evidence that is not the case. It is also surprising how different modes of cilia shedding and regeneration rates are not exactly correlative to recruitment of IFT components to the cilia base.

The manuscript is well written and data/analysis is rigorous. I only had minor comments that should be addressed.

Fig. 5B. I am confused or maybe there was some mistake in 5B (mechanical shear)? The data for cilia (blue circles) shown on the right suggest cilia are not lost after mechanical shear. However that is opposite of what you see in Fig 5A and C in the same timeframe. Please reconcile.

During mechanical shear, there is a ~1 μ m track of cilia maintained on the cells post shear (cilia do not completely sever. They break off most of the way, but cells maintain that track) so we consider them still ciliated, though they just have very short cilia immediately post shear. We do this because as soon as cilia are visible upon ciliogenesis (~1 μ m long), we count the cell as ciliated, so this is done for consistency. Starting at line 252, we have written in the manuscript, "*Looking more closely at regeneration in these processes, we found that mechanically sheared cilia, which maintain a ~1 μ m ciliary track immediately following ciliary excision, can immediately begin steadily regenerating overtime (Fig 5C).*"

Fig 5E,F. Looks like individual data points are in ~3 different colors. Is the coloring important or should there be a legend? As is, it is a little hard to see the mean/error bars.

We have added a description in the legend where it first appears (Figure 1C): "*Orange circles are trial 1, purple circles are trial 2, and navy circles are trial 3. Small circles represent individual data points and overlaid large circles indicate the mean for the trial.*" We have also added to the statistical analysis methods for superplots, "*Small circles are individual data points. Large circles are averages for each trial. Orange circles represent trial 1 data, purple circles represent trial 2 data, and navy circles represent trial 3 data.*"

We have doubled the thickness of the error bars.

Fig S1. This seems pretty important point to the paper and is even referred to twice in the discussion. Why not make it Fig. 6? It seems like an interesting but rather counter-intuitive point that cells that are slower to regenerate cilia are actually more rapidly/intensely recruiting IFT components. It was discussed reasonably but I felt this should be a little more front and center in the MS rather than supplemental.
We have changed this to figure 6.

Fig S2. I also felt like it should be a main figure. There are a lot conditions to digest and it may be helpful to the reader to have this up front for comparison rather than having to re-direct.
We have changed this to figure 7.

Line 125 - please define PtK1 cells.

We have made the following change: "...female rat kangaroo kidney epithelial cells (PtK1) cells..."

Reviewer #2 (Comments to the Authors (Required)):

Microtubules play an important role in ciliogenesis. While many researchers have indicated that microtubules are a key component of cilia, it is unknown how cytoplasmic microtubules contribute to ciliogenesis. The authors investigate the link between ciliogenesis and cytoplasmic microtubule regulation in *Chlamydomonas* using chemical (Paclitaxel) and mechanical disruption techniques. They discovered that (1) the availability of soluble tubulin via cytoplasmic microtubules is not essential for ciliogenesis, (2) cytoplasmic microtubule depolymerization is not required for cilia construction, and (3) rather intact cytoplasmic microtubules significantly facilitate ciliogenesis. This work is well-written, and overall the findings are convincing. I just have one question.

The observation and measurement of cytoplasmic microtubules would be critical in this article to validate the author's hypothesis. Except for Fig. 1A, it is unclear how the authors quantify β -tubulin. In Figs. 3 and 4, for example, how did the authors investigate cilia and cytoplasmic microtubules positive *Chlamydomonas*? Could authors simply differentiate (or measure) cilia+/CMTs- and/or cilia-/CMTs+?

We have quantitatively measured microtubules through beta tubulin labeling. We describe the polymerization state of the microtubules as either polymerized (beyond half the long axis of the cell) or depolymerized (less than half the long axis of the cell). This was a very clear observation; either microtubules were noted to be present at the apex of the cell (barely extending into the cell such as in the DMSO control in Figure 1E) or were clearly extending beyond half the long axis of the cell into the cell which always occurred very rapidly (within 10 min). These measurements made the comparison of the relationship between cilia/ciliogenesis and the presence of microtubules more clear.

We have added to the methods under B-tubulin immunofluorescence and quantification: "For quantification, max intensity projections were generated from z stacks and 100 cells per time point were counted using the Cell Counter tool in FIJI for polymerized or depolymerized CytoMTs. Cells considered to have polymerized CytoMTs were cells with fluorescent CytoMTs spanning more than half the long axis of the cell"

(ciliary apex to the cell base). Cells considered to have depolymerized CytoMTs were cells with fluorescent CytoMTs that spanned less than half the long axis of the cell. These typically barely extended into the cell from the ciliary apex. The presence or absence of cilia was noted alongside the CytoMT assessment per cell.”

In cilia+ cells, was cilia length comparable between cilia+/CMTs- cells? These must be clarified before publishing. In summary, this work will be useful for understanding Chlamydomonas ciliogenesis, and this information may also be critical to understanding ciliogenesis in animal cells.

This was an interesting question and we have added this measurement as Figure 3C to compare ciliary lengths between *fa2-1* mutants which maintain their cilia during and following pH shock. The ciliary lengths were comparable between cells with or without CytoMTs which suggests that CytoMT depolymerization in cells does not impact measurable ciliary length during these timepoints.

September 19, 2023

RE: Life Science Alliance Manuscript #LSA-2023-02287R

Dr. Prachee Avasthi
Geisel School of Medicine at Dartmouth College
74 College St
HB 7200
Hanover, NH 03755

Dear Dr. Avasthi,

Thank you for submitting your revised manuscript entitled "Determinants of cytoplasmic microtubule depolymerization during ciliogenesis in *Chlamydomonas*". We would be happy to publish your paper in Life Science Alliance pending final revisions necessary to meet our formatting guidelines.

- please note that the titles in the system and on the manuscript file must match
- please add an Author Contributions section to your main manuscript text
- please add a conflict of interest statement to your main manuscript text

A. FINAL FILES:

B. MANUSCRIPT ORGANIZATION AND FORMATTING:

****It is Life Science Alliance policy that if requested, original data images must be made available to the editors. Failure to provide**

original images upon request will result in unavoidable delays in publication. Please ensure that you have access to all original data images prior to final submission.**

The license to publish form must be signed before your manuscript can be sent to production. A link to the electronic license to publish form will be sent to the corresponding author only. Please take a moment to check your funder requirements.

Sincerely,

October 3, 2023

RE: Life Science Alliance Manuscript #LSA-2023-02287RR

Dr. Prachee Avasthi
Geisel School of Medicine at Dartmouth College
74 College St
HB 7200
Hanover, NH 03755

Dear Dr. Avasthi,

Thank you for submitting your Research Article entitled "Determinants of cytoplasmic microtubule depolymerization during ciliogenesis in *Chlamydomonas*". It is a pleasure to let you know that your manuscript is now accepted for publication in Life Science Alliance. Congratulations on this interesting work.

DISTRIBUTION OF MATERIALS:

Again, congratulations on a very nice paper. I hope you found the review process to be constructive and are pleased with how the manuscript was handled editorially. We look forward to future exciting submissions from your lab.

Sincerely,
